# Influence of Mask Wearing during COVID-19 Surge and Non-Surge Time Periods in Two K-12 Public School Districts in Georgia, USA

**DOI:** 10.3390/ijerph20095715

**Published:** 2023-05-04

**Authors:** Xiting Lin, Fatima Ali, Traci Leong, Mike Edelson, Samira Hampton, Zoey Zuo, Chaohua Li, Chris Rice, Fengxia Yan, Peter T. Baltrus, Sonya Randolph, Lilly Cheng Immergluck

**Affiliations:** 1Department of Microbiology/Biochemistry/Immunology and Clinical Research Center, Morehouse School of Medicine, Atlanta, GA 30310, USA; 2Department of Biostatistics & Bioinformatics, Rollins School of Public Health, Emory University, Atlanta, GA 30322, USA; 3InterDev, LLC., Roswell, GA 30076, USA; 4Department of Community Health and Preventive Medicine, Morehouse School of Medicine, Atlanta, GA 30310, USA

**Keywords:** COVID-19, school-age children, delta variant, mask

## Abstract

Background: Into the third year of the COVID-19 pandemic and the second year of in-person learning for many K-12 schools in the United States, the benefits of mitigation strategies in this setting are still unclear. We compare COVID-19 cases in school-aged children and adolescents between a school district with a mandatory mask-wearing policy to one with an optional mask-wearing policy, during and after the peak period of the Delta variant wave of infection. Methods: COVID-19 cases during the Delta variant wave (August 2021) and post the wave (October 2021) were obtained from public health records. Cases of K-12 students, stratified by grade level (elementary, middle, and high school) and school districts across two counties, were included in the statistical and spatial analyses. COVID-19 case rates were determined and spatially mapped. Regression was performed adjusting for specific covariates. Results: Mask-wearing was associated with lower COVID-19 cases during the peak Delta variant period; overall, regardless of the Delta variant period, higher COVID-19 rates were seen in older aged students. Conclusion: This study highlights the need for more layered prevention strategies and policies that take into consideration local community transmission levels, age of students, and vaccination coverage to ensure that students remain safe at school while optimizing their learning environment.

## 1. Introduction

Since the early phase of the novel coronavirus 2019 (COVID-19) global pandemic caused by a novel beta coronavirus, severe acute respiratory syndrome coronavirus 2 (SARS-CoV-2), more than 18% of the pediatric population in the United States (U.S.) has become infected [1,2]. This is likely an underreporting of children infected due to the limited requirement of reporting systems in place in this country, and approximately a third of children may have mild and asymptomatic infections [3]. Hospitalizations, and severe complications, including the multisystem inflammatory syndrome in children (MIS-C), from COVID-19 infections have steadily climbed in this vulnerable population, especially since the beginning of 2022, when the Omicron subvariant of concern caused a ‘surge’ of infections across the U.S. and worldwide [4]. Recently, the U.S. Centers for Disease Control & Prevention (CDC) reported an estimated COVID-19 seroprevalence of ~79.8% of children between the ages of 6 months to 17 years [5,6]. While we are still learning how infection leads to COVID-19-related severe complications, e.g., multisystem inflammatory syndrome in children (MIS-C) and post COVID-19 conditions, it has become increasingly clear that until sufficient numbers of children and their family members have been vaccinated and SARS-CoV-2 becomes an endemic pathogen, the risk of transmission in the school setting will remain a significant public health concern. Effective mitigation strategies to prevent transmission in this age group are more urgently needed, given the continued surges of variants and subvariants more than three years into this pandemic. Several studies have previously demonstrated that wearing masks has been shown to decrease the spread of SARS-CoV-2, given that the primary transmission route of the virus is through respiratory particles [7]. While studies have reported that children who wore masks often complained about difficulty breathing, several studies have demonstrated no significant differences in oxygen saturation, respiratory rate, or pulse rate were found compared to children who did not wear masks [8,9,10].

In the U.S., although children six months and older are eligible for COVID-19 vaccination, most children remain unvaccinated [11]. Unlike during the first year of the pandemic when many K-12 schools were primarily using remote learning, there is currently emphasis during the second and third years of the pandemic to resume in-person learning for K-12 schools without regard to community transmission rates. Rising case rates of infected children, SARS-CoV-2 variants and subvariants with increased transmissibility, and persistently low COVID-19 vaccination rates, especially among the younger aged children, contribute to why there is an even greater urgency to better understand what mitigation strategies will effectively prevent spread in school or childcare settings, where children spend most of the day with other children.

Early in the pandemic, studies demonstrated variable levels of household transmission to children [12,13]. In these studies, when the prototype or ancestral variant was the dominant circulating variant, the rates of infection in children 5–17 years old were similar to that in adults [12]. Estimates of the secondary attack rate of SARS-CoV-2 in households and the factors associated with transmission have been postulated to be related to close contact, lack of face coverings, and inadequate ventilation [12]. As SARS-CoV-2 continually mutates, studies demonstrated that the Delta variant was more transmissible in children and adolescents compared to adults [14,15]. As of 19 March 2021 (the pre-Delta wave), the U.S. Centers for Disease Control and Prevention recommended universal indoor masking and maintenance of at least three feet physical distance between students in classrooms, and to implement ‘layer multiple prevention strategies’ [16,17]; these general recommendations changed as the pandemic continued with policies governing prevention strategies falling largely on the local state government and not federal levels to implement. The Georgia Department of Education and Georgia Department of Public Health released the ‘Georgia’s Path to Recovery for K-12 Schools’ guidelines on 1 June 2020 and a revised version on 13 July 2020. Recommendations include wearing face coverings, using hand sanitizer, and disinfecting areas. These guidelines, however, were not required by the state, and implementation was up to individual school districts [18].

The degree of transmission of respiratory pathogens such as SARS-CoV-2 in children likely vary with the age of the child. In general, especially with airborne pathogens, e.g., *Mycobacterium tuberculosis*, younger aged children (<5 years) are less prone to transmit to others compared to older aged children or adults. This is due to the fact that younger aged children are generally less able to generate the forceful cough required to be infectious and therefore cause the secondary attack rate to be lower than that for older children and adults [19,20]. Moreover, there has yet to be a study that specifically addresses the age of children and the transmission risks of SARS-CoV-2 in the school setting, where school districts have different enforcement of mask-wearing policies during school hours by both students and staff.

With the recent SARS-CoV-2 variants Delta and Omicron, both more ‘infectious’ compared to previous variants, the effectiveness of layered prevention strategies (e.g., face covering, enhanced ventilation, handwashing, contact tracing in combination with quarantine and isolation, cleaning/disinfection, etc.) are hard to ascertain, especially given shortages of testing, too few dedicated staff to conduct contact tracing, and lack of resources to meaningfully enforce quarantine/isolation guidelines. It is clear that studies are needed to better understand the parameters that effectively limit transmission in a ‘real-world’ setting, despite these limitations.

We hypothesize that school districts with ‘strict’ mask mandates are able to reduce the risk of transmission compared to school districts where mask wearing is optional during periods when community transmission rates of the Delta variant were high. Furthermore, we also hypothesize that children attending elementary school (younger aged children) have lower transmission rates than children and adolescents attending middle- or high-schools, regardless of the mask wearing requirements. While there are other studies reporting the risk of transmission in the K-12 school setting [2,12], we are the first to report transmission rates based on the school age of the child, comparing elementary-aged students to those in middle school and high school, between two school districts within the same state in this part of the U.S., but with differing masking policies during the period where the predominant circulating COVID-19 variant was the Delta variant.

## 2. Materials and Methods

Study Area. A school district is a defined area containing schools grouped together and governed by the same local school board. In the state of Georgia (located in the southeastern part of the U.S.), counties are generally the geographic area in which school district borders are drawn. Exceptions are some larger cities, which may have their own independent school districts. Each school, determined by the governing board, will have a set boundary that determines which students will attend based upon their registered home address, which is called the ‘zone of attendance’. School attendance zones are established to distribute school-aged children population as evenly as possible [21].

Study Design. Two school districts in the state of Georgia (DeKalb County School District, DCSD, and Cobb County School District, CCSD) were compared for pediatric cases of COVID-19. These two school districts were selected as they are both large public-school districts in the urban city of Atlanta and its metropolitan statistical area (MSA). Moreover, each district (spatially in different areas) enforced differing mask-wearing policies during the time period of this analysis. The DCSD required students and teachers to wear masks when indoors and while riding school buses [22]. The CCSD had an optional mask-wearing policy for students and teachers [23]. Furthermore, two time periods (August 2021 and October 2021) were compared. August 2021 represented the majority of the Delta variant surge, when reported COVID-19 weekly cases in school-aged children in Georgia were high, and October 2021 represented the tail-end of the Delta surge, when the reported COVID-19 weekly cases in school-aged children were much lower [24]. We selected these two time periods due to the differences in community transmission at the time and as it relates to when the K-12 academic school year started for the state of Georgia (1 August). 

School Attendance Zone Assignment/COVID-19 Cases by School Attendance Zone. Data were obtained from the Georgia Public Health Information Portal for persons under investigation with positive COVID-19 test results [25]. Cases included in the analysis were school-aged children (4–18 years old) who had residential addresses in either DeKalb or Cobb County. Each case was assigned to a school level, which is defined as elementary (pre-K-5th grade, ES), middle (6th–8th grade, MS), and high school (9th–12th grade, HS), based on the date of birth and Georgia’s August 1st cut-off date for kindergarten (K) enrollment. Georeferencing of all data was performed using ArcGIS Pro 2.8 (ESRI, Redlands). Coordinates were projected to Georgia West State Plane, NAD83, feet (EPSG: 2240). Relevant boundaries included individual school zones of attendance boundaries for the DCSD and CCSD. For the CCSD, the zones of attendance were obtained as PDFs from the CCSD website [26] and then manually digitized in ArcGIS Pro version 2.8 [27]. For the DCSD, the zones of attendance were provided by the school district as shapefiles (.shp). The cases were spatially joined and overlaid with school attendance zones at each school level to determine the number of cases in each attendance zone. In zones of attendance with more than one school of the same school level (e.g., two elementary schools), cases and school enrollment for each school were summed. Schools that did not fall under the school levels defined above were excluded from the analysis. For example, a school for the 3rd to the 8th grade includes the ES and MS levels and therefore would not have been included in our analysis. Case rates (cases per 100,000 students) were calculated based on the number of cases in each attendance zone and school enrollment number. School enrollment numbers were obtained for fall 2021 from the Georgia Department of Education Data Reports (accessed on 11 November 2021) [28]. 

Spatial Mapping. Case rates were defined as the number of COVID-19 cases in a school zone of attendance divided by the total number of students in the school zone of attendance multiplied by 100,000. Case rates were mapped in increments of 250 cases per 100,000 students in ArcGIS Pro version 2.8 [27]. Twelve maps were created (two school districts: CCSD, DCSD; three school levels: ES, MS, and HS; and two time periods: COVID-19 surge period (August 2021) and non-surge period (October 2021)). A defined interval classification method for the data was used, including the minimum and maximum rates for both school districts. This was done in order to compare surge and non-surge periods directly. By doing so, the relative rates of transmission at these various times could be illustrated, which would not be possible if individual classification methods for each school district was applied. The City of Marietta operates its own independent school district separate from Cobb County, even though its borders lie within Cobb County and therefore, this district was not included in the maps for the CCSD. Similarly, the City of Decatur has its own independent school district separate from DeKalb County; hence, the City Schools of Decatur school district was excluded.

Statistical Analyses. Table 1 lists the variables used and the data sources. County measures included race and ethnicity (Black, White, Asian, Hispanic), population over three years old enrolled in a public school, and uninsured people; these variables were obtained from the U.S. Census Bureau [29]. District level variables included race and ethnicity (Black, White, Asian, Hispanic), number of schools in each school level, number of students within each school level, average school size, average student-to-teacher ratio, and average free or reduced lunch. Race, ethnicity, student-to-teacher ratio, and average free or reduced lunch data were obtained from the National Center of Education Statistics [30]. The chi-square test was used to compare the categorical county and school district measures, and t-tests for continuous variables for the DeKalb versus Cobb counties and school districts.

The odds ratio comparing the DCSD to CCSD will be estimated via the school level (ES, MS, and HS) and time period (August 2021, surge period, and October 2021, non-surge period). To evaluate the effect of covariates on the rate ratio of COVID-19 cases among K-12 students during the Delta surge/no surge periods and the respective school districts, a multivariable negative binomial regression was used due to the overdispersion observed among cases during the non-surge period (October 2021) and adjusted for the surge period (August 2021). Covariates include school district, surge period (August 2022) COVID-19 cases, ethnicity (Hispanic), and student-to-teacher ratio (evidence for crowding). Significance was set as *p* < 0.05. All statistical tests were performed in RStudio 2022.02.0.

## 3. Results

We included 209 school attendance zones/boundaries, which reflects 224 unique schools from the two school districts, DCSD and CCSD; we excluded 17 schools that crossed over more than one school attendance zone (Figure 1). The population demographic variables of the two counties were significantly different in all county measures, including race, Hispanic ethnicity, proportion of students in various levels of K-12 grades, and health insurance status (Table 2). DeKalb County had a higher percentage of African-Americans and Asian populations and a higher percentage of people uninsured than Cobb County. Cobb County had a higher percentage of White and Hispanic populations and children three years old and older who were enrolled in any K-12 public schools. Population characteristics for county level data differed from respective school district level data. Hispanic students made up a larger percentage of the DCSD than DeKalb County whereas White students made up a smaller percentage of the DCSD than DeKalb County. A similar observation was seen for Cobb County when compared to its school district. Both school districts had similar number of total schools and number of schools at each school level (elementary, middle, and high school), but the CCSD had significantly higher number of students in each school level; the average school size for elementary and high school, and student-to-teacher ratio was also higher in the CCSD. The DCSD had a significantly higher percentage of students receiving free or reduced lunch. The average and median incomes were higher in Cobb County ($116,708 USD and $88,029 USD respectively) compared to DeKalb ($104,428 USD and $70,985, respectively) [29].

There were more CCSD elementary schools with higher case rates (per 100,000 students) in August 2021 than the DCSD, whereas in October 2021, more CCSD elementary schools had lower case rates than the DCSD (Figure 2A). A similar trend was observed for middle schools (Figure 2B) and high schools (Figure 2C). The area with high case rates in the northeastern part of Cobb County predominantly has a higher income, White, single-family owner-occupied community. In contrast, for DeKalb County, the areas with high case rates are in the southeastern region, where there is a higher proportion of lower income families, and higher proportion of African-American/Black people. Regardless of school district and school level, the incidence rate ratio reflected a higher rate in a COVID-19 surge period (August 2021) compared to a non-surge period (October 2021). For the CCSD, the rate ratio was the highest for MS (3.55) compared to ES and HS (2.89 and 3.35, respectively) (Appendix A). For the DCSD, the incidence rate ratio was the highest for high schools (2.25) compared to ES and MS (2.03 and 2.17, respectively). During a COVID-19 surge, when there was high community transmission (August 2021), the DCSD had significantly lower odds than the CCSD for ‘overall’ school-aged children and specifically for ES (Table 3). However, during a period of lower community transmission (October 2021), the CCSD had significantly lower odds than the DCSD for ‘overall’ school-aged children and also across all levels of schools. Appendix A shows the community transmission rates during August 2021 and October 2021 from the Georgia Department of Public Health [33].

In the negative binomial regression (Table 4), October 2021 COVID-19 cases were adjusted for the following covariates: school district, Delta surge period COVID-19 cases, Hispanic ethnicity, and student–teacher ratio. As shown in Table 4, there was an 8.5% decrease in the incident rate of October cases for every unit increase in the student–teacher ratio. For every 100-student increase in Hispanic children, there was a 3.3% decrease in October cases. Students attending the CCSD were 0.7236 (95% CI 0.5997–0.8732) as likely to test positive for COVID-19 in October than the DCSD students with all other variables being equal (*p* = 0.0007).

## 4. Discussion

In our study, we demonstrate significant differences in COVID-19 cases between two school districts with quite different sociodemographic profiles. While the incidence rate in the school district with mandatory mask wearing was higher compared to the school district with optional mask wearing during the non-surge COVID-19 period, we did observe that overall, the age of the student (and the respective school level) did affect the risk for COVID-19. Although our study is based on the premise that schools adhered to their district’s guidelines on masking without a break in adherence, we did not conduct a sensitivity analysis with varying rates of adherence either with the masking or not masking rule. For example, in DeKalb County, students were allowed to forgo masking during athletic activities, and enforcement of these rules are likely less than optimal given that the state’s policies essentially discouraged mask wearing during this period in this setting. Our results are, therefore, a conservative estimate of the impact of masking on transmission rates. While we could not ascertain adherence in Cobb County either, a report of an entire fifth grade class in this mask optional district being sent home after a significant number of students were found to test positively for COVID-19 suggests that the ‘mask optional’ rule likely contributed to these types of situations [34].

During the Delta period of the pandemic, political party allegiance within a community may also have contributed to the acceptance of no mask wearing in schools and other public settings. Policies for mask wearing were determined at the state level. In the state of Georgia (a largely Republican state), the Republican governor signed an executive order during this period, which prevented public schools from using the state’s public health emergency status as a basis for mask mandates. Other states with Republican governors, e.g., Florida, went so far as to threaten to cut funds from school districts that broke rank with the statewide ban on classroom mask mandates [35].

Our findings are similar to what others have reported [14,15] for COVID-19 transmission among K-12 school aged children and the correlation with transmission in the community when community transmission levels are low, during a period when overall pediatric vaccination rates are also low. In contrast, when community transmission of SARS-CoV-2 is high, as was seen in August 2021 during the Delta variant wave, then strict adherence (not optional) to masking does make a noticeable impact in decreasing transmission among children in school settings or other settings where there are high numbers of children in closed space, e.g., daycare centers. While our study suggests that masking is not noticeably ‘preventing’ COVID-19 transmission when the amount of virus circulating is relatively low, this result further emphasizes the role for a layered prevention strategy. The ‘risk tolerance’ scale should be based in part on actual community transmission rates, when significant numbers of children (and their household members) remain vulnerable to infection.

Previous studies on mask-wearing in children and schools examined changes in transmission of COVID-19 cases in a period of time, such as spanning a month or several months that included surge and non-surge periods [36,37,38,39,40]. Other factors driving COVID-19 cases when community transmission rates are low may be also related to inherent features of the virus itself. With Delta, and now Omicron and its subvariants, there has been generally underreporting, given that the symptoms for many are subclinical or mild and many are ‘home testing’ and not reporting the diagnosis to their healthcare providers or public health authorities. The infectious nature of the child may also be affected by the age of the child, which has been seen with other respiratory pathogens including influenza. While most elementary aged children remain unvaccinated against COVID-19, many may have had COVID-19 exposure and therefore, have some level of immunity from prior infection, which may reduce their risk for transmitting when infected with another variant, such as the Delta variant during our study period. The Delta variant has been demonstrated to be more highly transmissible than the ancestral or prototype variant of SARS-CoV-2, and given that schools often are densely populated, it is not surprising that transmission rates were higher during a COVID-19 surge. With the Omicron variant and its subvariants demonstrating an even more transmissible capability than the Delta variant, developing targeted masking strategies remains important in settings where crowding occurs, especially when these settings involve children who remain a largely unvaccinated population in the U.S.

Regardless of the district masking policies, schools are not independent of their larger community, and will be affected not only by overall transmission rates, but also community characteristics and social behavioral norms in which they reside. For example, non-White race and older age have been associated with more mask wearing and increased testing behaviors [41]. Neighborhoods where a larger proportion of their residents are older (>65 years) are likely to have higher vaccination rates and reflect communities that exhibit more cautious social behaviors that lower their risk for infection, including wearing masks in crowded indoor settings. Schools located within these types of neighborhoods, in turn, may develop policies that are aligned with the community they are located in. While we did not examine the neighborhood level factors that may contribute to why certain zones of attendance were higher in case rates during the Delta wave compared to the adjacent school zones, mitigation behaviors apart from masking within specific schools of a zone may account for the ‘divergent’ rates of COVID-19 of these adjacent zones within a district.

The race and ethnic disparities seen with COVID-19 infection since the start of the pandemic have persisted at the state of Georgia and its county levels. However, our results show this may be seen in smaller geographic units (neighborhood level) [42,43]. We chose two districts that serve significantly different sociodemographic communities: the DCSD had a higher proportion of African-Americans and uninsured students whereas the CCSD had a higher proportions of White, Hispanic, and children over three years old who enrolled in public schools. Moreover, the DCSD has a higher percentage of school-aged children receiving free or reduced lunch. The CCSD had a higher number of enrolled students, higher average school size, and higher student-to-teacher ratio across all three school levels. These characteristics of the CCSD may contribute to the increased ‘density’ of the schools within its district and therefore increase the risk for transmission, especially if masks are not required during high COVID-19 transmission. In comparison, the DCSD is a district with a higher proportion of students who are from neighborhoods with a higher proportion of African-Americans; particularly in the far south end of the county, the neighborhoods may be up to 90% African-Americans. In our spatial analyses, we observed that regardless of the school level, higher case rates were seen in communities with large proportions of African-Americans for DeKalb County. DeKalb County also has more areas with higher case counts in lower income neighborhoods compared to Cobb County. In Cobb County, the southeast corner of the county is racially more diverse (mix of White, Black, Hispanic, and Asian) and has a higher renter occupancy, which may be a proxy for crowding and therefore, contribute to the higher incidence rates seen.

Mask-wearing policies in schools have been highly debated with concerns for its impact on socioemotional development and even, speech delays particularly in the younger aged school children. In contrast, mask-wearing in children in daycare centers nationwide reduced daycare closures by 13% [44]. Moreover, still much remains unknown in terms of the longer-term consequences of natural COVID-19 infection, particularly in the areas of neuro- and cardiovascular diseases. The rates of post-COVID-19 continue to rise, especially among the pediatric population. All these issues must be factored when determining the safest strategy to implement in schools, particularly those where the socioenvironmental setting is not optimal to prevent the spread of a rapidly changing virus, e.g., smaller classroom space with suboptimal ventilation. This is relevant even as we move out of the pandemic state to the endemic state.

The low vaccination rates for these two districts also may contribute to the findings seen during October 2021, a period of low community transmission. In Dekalb County, the vaccination rates overall were 27.6% (end of August 2021) and 32.3% (end of October 2021) for fully vaccinated, and 32.8% (end of August 2021) and 38.3% (end of October 2021) for people 12 years old and older. In comparison, Cobb County had higher vaccination rates. Overall, 34.4% (end of August 2021) and 39.2% (end of October 2021) were fully vaccinated. For children 12 years old and older, 40.5% (end of August 2021) and 46.2% (end of October 2021) were fully vaccinated. 

Limitations: Data from the Georgia Department of Public Health showed a more uniform data collection and reporting system than school-reported numbers, whose methods and contact tracing may differ among school districts. This data source would also allow greater scalability of this study to encompass all public-school districts in Georgia. Using this data has some limitations. It was assumed that children started school based on the date of birth cut-offs established by the Georgia Department of Education. However, there are several potential reasons, such as moving from another state or parents’ preference to wait, where a child may not follow these established cut-offs. Additionally, it was assumed that all school-aged children attended a public school since it is unknown whether these children attend private school or home school. Students may also attend a public school outside of a school’s zone of attendance. Consequently, the magnitude of the case rates may be under- or overestimated. However, the ratios of COVID-19 cases are comparable and worth noting to tell this interesting story.

We were not able to ascertain the household settings of the students who attended the various schools of these two districts. It has been demonstrated that household transmissions and the risk for infectivity differ based on the age of household members among other parameters (vaccination status of household members). Findings from this study cannot be generalizable to other communities in states that had differing policies around masking during this period of the pandemic. We also could not determine the level of adherence to school district policies around masking or other mitigation strategies, which are strongly influenced by the adult educators and school staff. Because the act of wearing a face mask during the time of this study has been aligned with political affiliation, we could not assess how this impacted behavior within each district; we do acknowledge that these districts represented different political leanings, which may affect adherence of both students and staff to mask wearing. Future studies of transmission patterns may involve prospective surveillance of cases’ household conditions and identify risk factors that occur outside of the school setting. Vaccination rates and proportions of natural infections of both teacher and staff are also areas worth exploring in future studies as we move through this pandemic and determine what degree of mitigation is needed to prevent transmission patterns that may lead to outbreaks. What defines high/low community transmission to have the greatest benefit/risk ratio is not yet well understood, but it is an important question to investigate further and would be the next logical step to take, as many parts of the nation drop mandatory mask-wearing in schools. Furthermore, here, two school districts from Georgia were used in the analyses, but it is important to increase the number of school districts and expand geographically.

## 5. Conclusions

This study highlights the need for more layered prevention strategies, and policies that take into consideration community transmission levels, age of students, and vaccination coverage to ensure that students remain safe at school while optimizing their learning environment. Mask wearing makes a difference on COVID-19 cases when community transmission is high during the Delta surge. However, during the period of low community transmissions of the Delta variant, optional mask wearing had no disadvantage compared to mandatory mask wearing in reducing transmission and case burden from the Delta variant.

## Figures and Tables

**Figure 1 ijerph-20-05715-f001:**
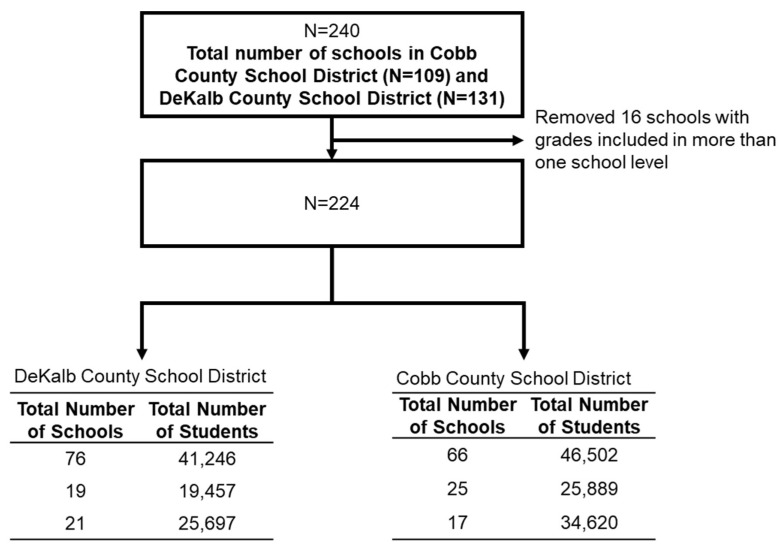
Enrollment scheme.

**Figure 2 ijerph-20-05715-f002:**
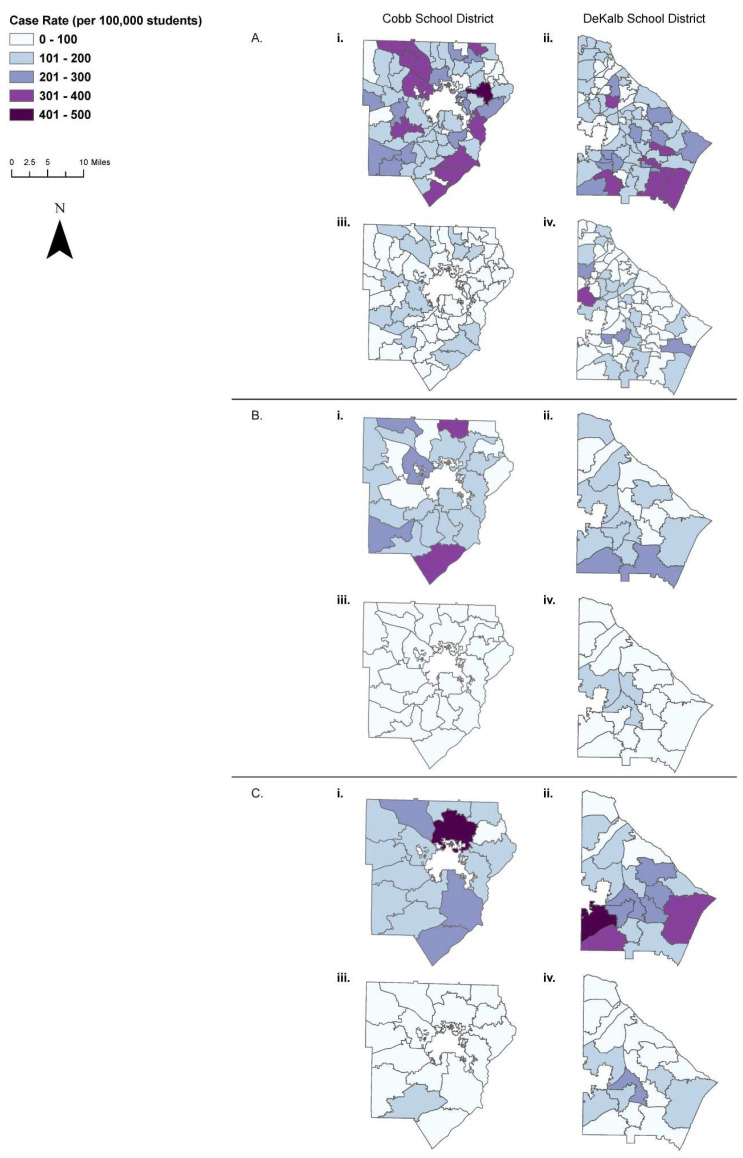
COVID-19 case rates in elementary (**A**), middle schools (**B**), and high schools (**C**) of two counties in two time periods: During the Delta-variant surge (August 2021) for the Cobb County School District (**Ai**,**Bi**,**Ci**) and DeKalb County School District (**Aii**,**Bii**,**Cii**); non-surge period (October 2021) for the Cobb County School District (**Aiii**,**Biii**,**Ciii**) and DeKalb County School District (**Aiv**,**Biv**,**Civ**). Case rates are reported based on cases of COVID-19 among students per 100,000 students who are within the respective zone of attendance. (This rate was calculated based on the number of COVID-19 cases in a school zone of attendance divided by the total number of students in the school zone of attendance multiplied by 100,000).

**Table 1 ijerph-20-05715-t001:** Summary of Data Source(s) for Area-based Population Characteristics.

Area Level Category	Variable	Source
County	Race: Black, White, Asian Ethnicity: Hispanic Population 3 years and over enrolled in school (Kindergarten to 12th grade) Uninsured	ACS 1-Year Estimates 2019 ^1^
School	Total schools Total students Average school size	Georgia Department of Education ^2^
	Race and ethnicity Average student–teacher ratio Average free lunch Race	National Center of Education Statistics 2019–2020 ^3^

^1^ ACS: American Community Survey Data-U.S. Census Bureau. Estimates from 2019 data [29]. ^2^ For fiscal year 2022-1 (5 October 2021). ^3^ ref [30].

**Table 2 ijerph-20-05715-t002:** Population Characteristics for Counties and School Districts.

Area Level Characteristic	Mask Required DeKalb County	Mask Optional Cobb County	*p*-Value ^1^
**County Level**			
Race and Ethnicity, N (%)			
Black	409,261 (52.6%)	212,839 (27.2%)	<0.001
White	256,642 (33.0%)	427,199 (54.6%)	<0.001
Asian	47,162 (6.1%)	41,478 (5.3%)	<0.001
Hispanic	64,540 (8.3%)	101,099 (12.9%)	<0.001
Population ≥ 3 years enrolled in school (Kindergarten to 12th grade), N (%)	121,974 (15.7%)	128,267 (16.4%)	<0.001
Uninsured	107,654 (13.8%)	97,012 (12.4%)	<0.001
**District Level**			
Race and Ethnicity, N (%)			
Black	55,392 (59.6%)	33,514 (30.1%)	<0.001
White	10,027 (10.8%)	40,910 (36.8%)	<0.001
Asian	6653 (7.2%)	6562 (5.9%)	<0.001
Hispanic	18,353 (19.8%)	25,441 (22.9%)	<0.001
Total schools, N (%)			
Elementary school	76 (65.5%)	66 (61.1%)	0.586
Middle school	19 (16.4%)	25 (23.1%)	0.269
High school	21 (18.1%)	17 (15.7%)	0.770
Total students, N (%)			
Elementary school	41,246 (47.7%)	46,502 (43.5%)	<0.001
Middle school	19,457 (22.5%)	25,889 (24.2%)	<0.001
High school	25,697 (29.7%)	34,620 (32.3%)	<0.001
Average school size, mean (std)			
Elementary school	543 (168)	705 (209)	<0.001
Middle school	1024 (282)	1036 (256)	0.888
High school	1224 (482)	2036 (581)	<0.001
Average student–teacher ratio, mean (std)			
Elementary school	14 (2)	13 (2)	0.119
Middle school	16 (1)	17 (1)	0.034
High school	16 (2)	18 (2)	0.001
Average free lunch percentage, mean (std)			
Elementary school	76 (32)	39 (29)	<0.001
Middle school	78 (26)	36 (26)	<0.001
High school	66 (25)	29 (21)	<0.001
Ventilation System (HVAC)	Upgraded	Upgraded	
Contact tracing performed	Yes [31]	Yes [32]	

^1^ *p*-value is statistically significant (*p*-value < 0.05). Chi-squared tests were performed for categorical variables comparing one level to the other levels (e.g., Black race vs other races). *t*-tests were performed for continuous variables. (std = standard deviation; HVAC = Heating, Ventilation, Air Conditioning).

**Table 3 ijerph-20-05715-t003:** COVID-19 Odds Ratio for the Mask Required School District to the Mask Optional School District in the Surge (August 2021) and Non-Surge (October 2021) Periods.

Time Period	School Level	Odds Ratio	95% Confidence Interval (LL, UL) *	*p*-Value
Surge Period (August 2021)	Overall	0.897	(0.837, 0.962)	0.0024
	Elementary School	0.847	(0.765, 0.938)	0.0014
	Middle School	0.860	(0.725, 1.022)	0.0845
	High School	0.990	(0.881, 1.113)	0.8659
Non-Surge Period (October 2021)	Overall	1.306	(1.167, 1.462)	<0.001
	Elementary School	1.188	(1.014, 1.392)	0.0335
	Middle School	1.356	(1.024, 1.796)	0.0334
	High School	1.478	(1.216, 1.795)	<0.001

* LL = lower limit, UL = upper limit.

**Table 4 ijerph-20-05715-t004:** Regression Model for the Non-Surge (October 2021) Period.

Variable	Incident Rate Ratio	95% Confidence Interval	*p*-Value
Hispanic (per 100 student increase)	0.9671	0.9358, 0.9995	0.0466
Student–teacher ratio	0.9151	0.8752, 0.9568	<0.0001
Surge Period COVID-19 Cases ^1^	1.0133	1.0056, 1.0211	0.0007
District (Mask Optional vs. Mask Required) ^2^	0.7236	0.5997, 0.8732	0.0007

^1^ Surge cases are defined as those occurring during August 2021. ^2^ Mask Optional School District is the Cobb County School District, and Mask Required School District is the DeKalb County School District.

## Data Availability

Restrictions apply to the availability of these data. Data were obtained through an agreement between the Georgia Department of Public Health and the National Center for Primary Care at Morehouse School of Medicine. Requests for data should be made to the Georgia Department of Public Health.

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
