# Peer review of "Influence of Mask Wearing during COVID-19 Surge and Non-Surge Time Periods in Two K-12 Public School Districts in Georgia, USA"

_ijerph, 2023, doi:10.3390/ijerph20095715_

Round 1

Reviewer 1 Report

Overall Comments

Thank you for the opportunity to review the manuscript “Influence of Mask Wearing During a CoVID-19 Surge and Non-Surge Time Period in Two K-12 Public School Districts in Georgia”. This is an interesting publication that adds to the body of knowledge about effectiveness of COVID-19 prevention strategies in school-aged children. The manuscript is generally well-written but would benefit from being run through a grammar check program or from careful review by the authors. The publication will also benefit from some content revision as described below.

Introduction Section

1. Line 35 “more of the pediatric population have become infected [1,2]” – Please clarify what you mean by this statement. Are you referring to cumulative number of infections since the start of the pandemic? If that’s the case then I don’t think that the citations provided are appropriate for the statement. Rather, there should be data on cumulative infections that can be referenced with an accession date to provide the reader with context of the state of the pandemic at the time the article was being written.

2. Lines 40 and 41 – “…estimated CoVID-19 seroprevalence of ~79.8% of children between the ages of 6 months to 17 years [4,5]”. The link for this citation is not working. Please provide a new citation that is more current.

3. Lines 44 and 45 – The majority of children in the US are eligible for Covid-19 vaccines. Please revise or remove this statement in light of that information.

4. Lines 48 and 49 – “…continued surges of variants and subvariants more than two years into this pandemic”. Please update this as we are more than 3 years into the pandemic now.

5. Lines 60 and 61 – “These studies, when prototype or ancestral variant was the dominant circulating variant, rates of infection in children 5-17 years old were similar to adults 61 [7]”. This sentence needs to be edited.

Material and Methods

6. Lines 116-117 – “when reported CoVID-19 weekly cases in school-aged children were much lower [18]”. The reference yields a school report from April 21, 2022. Is there a more appropriate reference that can be used to show the difference in incidence between the two time periods being compared in the study?

7. Lines 215 and 216 – “the area with high case rates are located in the southeastern region” should say “the areas with high case rates are…” or “the area with high case rates is…”.

Discussion

8. Lines 264 and 265 – “we did not conduct a sensitivity analyses with varying rates of adherence either with masking or not masking rule”. Can you please rephrase this? Having a mask option policy is different than having a “not masking rule”. Do you have any data on rates of mask wearing in Cobb County during the two study periods? I know that may not be available but it would be helpful to either present that or to address the fact that it is possible that students in Cobb County were voluntarily masking, potentially at a level that was also protective for cases of CoVID-19.  

Reviewer 2 Report

Congratulates the authors on the idea for the work and on writing the paper. I have the following minor comments.

L1 – ‘’ iArticle’’ - remove ""i"".

L30 – change ‘’ masking’’  on ‘’mask’’.

L39 – ‘’ U.S.’’ - This is a form of abbreviation, I suggest to elaborate.

L43 – ‘’ ))’’ - Remove the double parenthesis.

L41- ‘’ [4,5].’’ And L82 ‘’ [14,15].’’ - Why are the dots at the end of the sentence in red?

L80 and 94-95 - The use of italics is not necessary.

1. Introduction - The introduction is written correctly, however, I have one major comment on this part.

The title of the authors' paper is ‘’ Influence of Mask Wearing During a CoVID-19 Surge and Non-Surge Time Period in Two K-12 Public School Districts in Georgia’’. However, the authors do not address the effects of masks on the human body in the introduction. I suggest adding a paragraph or at least a couple of sentences about the effects of masks on the spread of the virus (DOI: 10.1073/pnas.2014564118 ), the respiratory system (doi: 10.1017/ice.2021.470), the cardiovascular system (DOI: 10.1513/AnnalsATS.202008-990CME ) and the musculoskeletal system (10.3390/jcm11020303). This will increase the interest and citability of the paper.

L14 – ‘’ ES, MS, and HS’’ - No matter the obvious abbreviations, if they are used for the first time they should be developed.

The results and discussion and conclusions in my opinion are written correctly and clearly.

Reviewer 3 Report

The study aims to find out the effect of mask wearing during surge and non-surge period across two school districts in Georgia. The study intended to use a scientifically sound methodology for the objectives it aimed to achieve. However, there are certain areas that need improvement, especially in Methodology as well as Results part.
Authors should add a sub-heading of "study area" and clearly write down details on the term School Districts. How many school districts are there in total, how many schools and students are there, what was the selection criteria and how many of the schools were selected or not on basis of what criteria. 
Yes, we can find some of the above-mentioned information from the text at different places but that needs to be placed at one place in a simplified manner for better understanding of the readers. And this information should better be reflected through a flow-sheet diagram for clarity. 
Lines 137-8 "schools having overlapped school levels 3rd to 8th grade" this is confusing and needs to be addressed.
Table 1 is very confusing, why all three columns are labelled as same? It should be simplified
In table 2, (std) stands for standard deviation? Please spell it out in the foot notes. Also, better to add a demarcation line in the table where N (%) ends and mean (std) is started. This will address the confusion.
In Figure 1and 2, the distance scale and north arrow be added. Also, labelling can be done inside the picture for A, B, C and D maps for better visualizing and understanding of readers. Also, bringing the Figure S1 to the main manuscript will be a good idea. Authors can label all three figures as Figure 1a, 1b and 1c respectively.
Lines 248-50 "There was an 8.5% decrease in the incident rate of October cases for every unit increase in student-teacher ratio. For every 100-student increase in Hispanic children, there was a 3.3% decrease in October cases", this data represents which table in the manuscript?

Can we replace rate ratio in table 3 with odd's ratio and explain it accordingly?. The current explanation doesn't seem to be convincing

In Table 4, Poisson regression model does not reflect complete information, in my opinion, intercept and standard error should be mentioned there. What is the explanation to this?
The heading Institutional Review Board Statement is there but it reflects no IRB approval letter number nor it reflects the name of the approval authority.

Round 2

Reviewer 3 Report

Authors have addressed all the comments professionally making the final product much improved.